# Training in Roller Speed Skating: Proposal of Surface Electromyography and Kinematics Data for Educational Purposes in Junior and Senior Athletes

**DOI:** 10.3390/s24237617

**Published:** 2024-11-28

**Authors:** Giulia Bongiorno, Giulio Sisti, Helena Biancuzzi, Francesca Dal Mas, Francesco Giuseppe Minisini, Luca Miceli

**Affiliations:** 1Friuli Riabilitazione Rehabilitation Center, 33080 Roveredo in Piano, Italy; 2Pain Medicine, IRCCS CRO National Cancer Center of Aviano, 33081 Aviano, Italy; 3Department of Economics, Ca’ Foscari University, 30121 Venice, Italy; 4Venice School of Management, Ca’ Foscari University, 30121 Venice, Italy; francesca.dalmas@unive.it; 5Collegium Medicum, University of Social Sciences, 90-229 Lodz, Poland; 6Department of Physics, University of Milan, 20133 Milan, Italy

**Keywords:** roller speed skate, surface electromyography, educational

## Abstract

**Introduction**: Roller skating shares biomechanical similarities with other sports, but specific studies on speed skaters are limited. Injuries, particularly to the groin, are frequent and related to acute and chronic muscle stress. Technology, particularly surface electromyography, can now be used to monitor performance and prevent injuries, especially those caused by muscular asymmetries. Such studies can be used to enhance training and for educational purposes. **Materials and Methods**: This pilot study was conducted on three subjects: two cadet-athletes and a novice, compared with the performance model of an elite athlete. Surface electromyography and kinematic analysis monitored the lower limb muscles during the propulsion and recovery phases of skating. Electrodes were placed on specific muscles, and triaxial accelerometers were used to detect kinematic differences and asymmetries. The results: Cadet 1 was closest to the elite athlete’s performance model compared to Cadet 2, especially in kinematics and muscle efficiency. However, both cadets showed electromyographic differences compared to the elite athlete, with uneven muscle co-activations. The novice exhibited more oscillations and earlier propulsion compared to the elite athlete. **Discussion**: Using electromyography and kinematic analysis made it possible to identify differences between elite athletes, cadets, and novices. These observations provide useful data for developing personalized training and educational plans and preventing injuries related to muscle overload.

## 1. Introduction

The individual movements of a roller skater show similarities with the biomechanics of hockey players and ice skaters [1,2], with intrinsic specificities to the discipline. The literature has already considered the aspect of kinematic and surface electromyographic analysis in speed roller skating [3]. Still, no study has yet been conducted on comparing a small cohort of subjects to create personalized training. In hockey, about 10% of injuries can be attributed to groin issues [4,5] in terms of acute or chronic stress on the adductor muscles, about the electromyographic peak reached with increasing speed. In other cases, stress injuries (recto-adductor syndrome) may occur in such athletes, compromising performance efficiency and, thus, their careers [6]. It has been found that during the competitive season, 20% of athletes sustain a non-traumatic injury, affecting their seasonal performance [7,8]. The multiplanar kinematic pattern, constrained during forward movement at the hip level, forces the athlete to find the best skate position because of the accelerations along the three spatial axes, and these stresses impact the activity of the lower limb muscle groups. When directly compared to running, the propulsive motor pattern of the skater exhibits greater lateralization of the entire lower limb [9,10]. Of fundamental importance, based on the kinematic acquisitions and electromyographic readings, is evaluating muscle activity, not only as an indicator of stress but also because of propulsive efficiency, which differentiates athletes in terms of performance. It has been hypothesized that excessive muscle tension, particularly in the adductor region, may be caused by eccentric contractions (explosive abduction during the stance) that attempt to decelerate the limb during lateral movement, followed by rapid concentric activation to facilitate maximum propulsion [11,12]. An asymmetry in terms of the electromyographic peak between the right and left adductor (reduced impulse by 80%) and the co-activation ratio with the gluteus medius in terms of muscle tension may increase the susceptibility of the long adductor to injury risk [13]. The use of technology is widely recognized as a key tool for performance enhancement, prevention, and rehabilitation in both the sports and clinical worlds. These technologies offer a range of new opportunities for quantitative assessment and medium- to long-term movement monitoring [14]. The surface electromyographic algorithm examined, as described below, will allow professionals to monitor the skating pattern throughout the entire push and recovery cycle, obtaining valuable data, not only to observe the cyclicality of the motion (e.g., whether it is consistent during each propulsion and recovery step) but also to highlight the electromyographic peak with the corresponding percentages of muscle involvement and co-activations between agonist and antagonist muscles [15]. The electrodes can capture signals from the motor unit and, therefore, from all the fibers it innervates [16]. The use of sensors has allowed biomechanics to be monitored by comparing different sampled populations, i.e., between male and female athletes, enabling the identification of possible technical variations between men and women that may affect performance as well as mechanical load [17]. These technical differences are an important point that the coach must consider in training physical preparation, given the training, performance, and anthropometric variables between the two sexes. With the analysis of the algorithm described below, the aim of this work is to canalyze data from electromyographic and triaxial accelerometric patterns between a novice and two cadets, compared with a professional athlete to offer, in the near future, a unique training method for athletes approaching competitive online racing skating.

Following these premises, this study aims to choose a set of kinematic and electromyographic parameters to guide the technical and athletic training of roller speed skaters. Such parameters may be valuable for both educational purposes and injury prevention. 

This study presents some innovative aspects compared to the literature, summarized above. Regarding the electromyographic part, the muscular co-activations were analyzed both in a punctual manner (Rudolph index) and in a dynamic manner, representing their value in graphic form during the entire skating cycle. Therefore, an analysis tool was created that can quantify and precisely locate the variations in co-activations within the two phases. Such an instrument constitutes the studied cyclic movement, with potential advantages for training and for the prevention of injuries to athletes. Furthermore, attention was paid to the asymmetries in the co-activations between the right and left lower limb, since roller speed skating is known to be an asymmetric sport. From the kinematic point of view, the athlete’s acceleration was analyzed, breaking it down into the three Cartesian axes, offering a potential tool for evaluating the effectiveness of the gesture to be explored in future studies.

## 2. Materials and Methods

This work is a pilot experimental study conducted at the municipal skating ring in Pordenone (Italy), in collaboration with the “*ASD Comina Skating Club*” sports association. Starting from the reference electromyographic performance model derived from a female athlete, a former world champion in speed roller skating, an electromyographic and kinematic analysis protocol for both lower limbs during linear movement was structured [3]. The study protocol allows for analyzing the phases of the skating cycle in terms of duration, muscle activation of the lower limbs, the degree of co-activation between agonist and antagonist muscles, and accelerations on the three body planes during the propulsion and recovery phases. This protocol is also able to detect differences between the two limbs (asymmetries). Three subjects were recruited and investigated: two 13-year-old right-handed female athletes, both active in competitions, and a 49-year-old senior novice, left-handed, who was new to speed roller skating. Participants voluntarily adhered to the study protocol, with the only inclusion criterion being the absence of underlying pathologies. The performance model concerned a 31-year-old elite female athlete, an Italian, European, and world champion in speed roller skating, whose data had already been published [3].

Kinematic evaluations were carried out during skating: for the athletes, before preparation for the competitive season, and for the novice, as soon as minimal stability on skates was achieved. Surface EMG was used to measure muscle activity during the skating test (BTS Bioengineering—Garbagnate Milanese, Milan, Italy—EMG “freemg 1000”, with a sampling frequency of 1000 Hz). The electrodes (24 mm diameter, Kendall Arbo^®^, Cardinal Health, Inc., Dublin, OH, USA) were applied to predetermined muscles of the right and left leg, following the guidelines of the Seniam project [18]. The muscles analyzed were the soleus, gluteus maximus, gluteus medius, adductor magnus, rectus femoris, biceps femoris, vastus lateralis, and tibialis anterior. The skin at the electrode application site was shaved, lightly rubbed with sandpaper, and cleaned with alcohol. The electrodes were secured with adhesive tape. At this point, each participant performed a maximal isometric contraction for three seconds for each muscle to collect EMG data for signal amplitude normalization. The EMG signals from the electromyograph at 1000 Hz were full-wave rectified, and the amplitudes were rescaled to maximum voluntary contractions: MVICs with 3-second contractions were performed. The time series were low-pass filtered using a 4th-order Butterworth algorithm with zero lag at a cutoff frequency of 10 Hz.

A triaxial accelerometer (200 Hz, G-sensor, BTS Bioengineering, Corp., Garbagnate Milanese, Italy) was placed at the level of the first sacral vertebra (S1) of the subject, secured using the adjustable strap provided with the instrument. A high-definition camera synchronized with the electromyograph and inertial sensor, recording at 50 frames per second (Vixta 50, BTS Bioengineering, Corp., Garbagnate Milanese, Italy), was positioned on the sagittal plane to capture the subject during both the straight and return phases. The camera was synchronized with both the inertial sensor and the electromyographic probes. The tests were conducted on an 80-meter straight, first investigating the right side and then the left side of each athlete, collecting telemetry wirelessly immediately after each acquisition. The data collected from the athletes included the following: EMG signals in absolute terms and as percentages relative to the specific MVIC (maximum voluntary isometric contraction) for each muscle described above (for the acquisition method, refer to the specific published work) during straight-line skating over approximately 80 meters (averaging 4 skating cycles); average acceleration values on the three spatial axes (X, Y, Z) during the propulsion and recovery phases of each skate (expressed as an average of the values obtained in the 4 cycles of analysis); Rudolph’s co-activation index of agonist-antagonist muscles and its graphical representation; skating duration (propulsion and recovery phases, expressed in milliseconds and as a percentage of the cycle duration).

## 3. The Results

The focal point of the analysis is to consider whether and how closely the subjects in training approached the already established performance model in the literature.

### 3.1. Elite Athlete—Cadets

#### 3.1.1. Kinematic Analysis Athlete—Cadets

The kinematic comparison represents the cornerstone on which the technical control of skating is based. As seen in the images (Figure 1a–c), the reference values relate to the percentages of the skating cycle and the average accelerations on the three axes. From these results, it can be stated that cadet 1 is technically closer to the performance model.

#### 3.1.2. EMG Analysis

A differentiated aspect between the model and cadet 1 is the ability to efficiently use the gluteus medius muscle: it can be observed that in cadet 1, the percentage of muscle usage in this area is different both in propulsion (cadet 1:30.5% versus 47.2%) and in recovery (cadet 1:22.6% versus 34.1%), unlike cadet 2, who shows greater graphical similarities and numerical findings in muscle usage ability (cadet 2: 42.3% versus elite athlete: 47.2%) when compared to the elite athlete (Figure 2a–c). The adductor longus in cadet 1 shows a muscle recruitment value in propulsion (elite athlete: 11.3% versus cadet 1:11.1%) and recovery (elite athlete: 17.3% versus cadet 1: 22.2%) that are closer to the elite athlete’s recruitment capacity. In contrast, cadet 2 presents propulsion values (elite athlete: 11.3% versus cadet 2: 14.9%) and recovery (elite athlete: 17.3% versus cadet 2:27.7%) that are slightly further from the performance model.

The figure indicates the sEMG value during the skating cycle phase of the right gluteus medius and right adductor longus muscles in the three subjects studied; the table below each graph indicates, during the propulsion and recovery phase (separated by the dotted vertical line), the peak electromyographic value, the percentage of the cycle at which this peak develops, the average electromyographic value, and its percentage distribution in the two phases.

#### 3.1.3. Co-Activation Analysis

In the first phase of skating, cadet 2 showed alternating co-activation percentages with ascending and descending phases, presenting an average close to the performance model (cadet 2: 25.4% versus 24.6%). Similar analysis is evident in cadet 1, where an initial peak and a sharp decline at the completion of the propulsion phase are observed, with lower co-activation indices (cadet 1: 11.6% versus 24.6%) (Figure 3a–c).

### 3.2. Elite Athlete—Novice

#### 3.2.1. Kinematic Analysis Athlete—Novice

It is immediately noticeable how the percentage values of the cycle are different, with the novice tending to anticipate the propulsion action (novice: 46.6% versus 55.5%), consequently increasing the recovery action (novice: 53.4% versus 44.5%). Another consideration must be given to the average accelerations, where the novice tends to show more oscillations along the three axes, creating excessive trunk movements during skating (propulsion: 6.568 m/s^2^ versus 8.557 m/s^2^, recovery: 5.548 m/s^2^ versus 9.644 m/s^2^) (Figure 4a,b).

#### 3.2.2. EMG and Co-Activation Analysis

The gluteus medius activity comparison between the elite athlete and novice shows discrepancies in terms of muscle activation percentage (propulsion: elite athlete 47.2% versus 34.4%; recovery: elite athlete 34.1% versus 26.2%), with different electromyographic impulse peaks between the two subjects, both in propulsion (elite athlete: 154.9 µV versus novice: 73.9 µV) and recovery (elite athlete: 118.1 µV versus novice: 41.1 µV). Adductor activation seems to show similar findings during propulsion in terms of muscle involvement (elite athlete: 11.3% versus novice 12.1%), with differences during recovery (elite athlete: 17.3% versus novice: 25.9%) and similarities in electromyographic peaks during propulsion (elite athlete: 30.4 µV versus novice: 37.9 µV), except for the recovery phase (elite athlete: 23.9 µV versus novice: 41.8 µV). Regarding the co-activation index, despite the numerical similarity in the recovery phase (elite athlete: 13.9% versus novice: 14.7%) and reduced co-activation during the propulsion phase (elite athlete: 24.6% versus novice 11.5%), the graphical trend reflects excessive value increases and decreases (Figure 5a,b and Figure 6a,b).

### 3.3. Asymmetry 

#### 3.3.1. Cadet 1 Asymmetry Index

It is noteworthy how the co-activation differences between the right side (14.4 versus 18.7) and the left side (20.8 versus 28.4) show percentage differences for the vastus lateralis and biceps femoris (Table 1). Regarding co-activation between the gluteus medius and adductor longus, good symmetry is recorded between the two body sides (left propulsion 29.6 versus recovery 19.1; right propulsion 29.1 versus recovery 22.9), particularly in the second phase of the cycle, where the left side shows less marked values but with a wide rise in the first phase. The results are reported in Table 1. 

#### 3.3.2. Cadet 2 Asymmetry Index

The biceps femoris analysis shows more marked values on the right side but with differences that could result in a noticeable and quantifiable asymmetry (propulsion: right 36.6 µV, left 30.8 µV; recovery: right 33 µV, left 32.3 µV). The vastus lateralis analysis shows predominance on the left side with doubled indices compared to the contralateral side (propulsion: right 90.4 µV versus left 194 µV; recovery: right 128.1 µV versus left 212.9 µV), indicating high asymmetry between the two regions. However, co-activation values are mostly consistent between the two sides, except for the right side’s propulsion phase (propulsion: right 33.9% versus left 26.4%; recovery: 17.5% versus 20.6%), where a major increase is noted. Of fundamental importance is the evaluation of the gluteus medius and adductor, as it can be observed that the major signals come from the left side, again presenting a marked asymmetry in the propulsion phase of co-activation values between the two sides in favor of the left side (right 24.2% versus left 33.3%). The results are reported in Table 2.

#### 3.3.3. Novice Asymmetry Index

High asymmetry is noted in the right-side peak of the vastus lateralis (propulsion: right 378.3 µV versus left 74.3; recovery: right 398.7 versus left 54.9). Regarding the biceps femoris, the asymmetries are less marked and noticeable (propulsion: right 19.7 µV, left 19.8 µV; recovery: right 28.4 µV versus left 17.8 µV). As for co-activation, there is a partial difference between the two regions in neuromuscular coordination (right 13 versus 16.2; left 17.7 versus 12). The gluteus medius shows more marked peak values compared to the contralateral side in the propulsion phase (right 27.3 µV versus left 13.2 µV) but smaller in the recovery phase (right 10.2 µV versus left 9.9 µV). As for the adductor longus, the evaluation presents contrary phenomena, with more pronounced findings on the left side (propulsion: right 41.7 µV versus left 45.8 µV; recovery: right 38.8 µV versus left 53.2 µV). As previously mentioned, the co-activation value to consider is represented by the propulsion phase of the right region (propulsion: 23.8% versus 6.6%). The results are reported in Table 3.

## 4. Discussion

### 4.1. Kinematics Comparisons

The world of sports is increasingly relying on technology to monitor its athletes comprehensively across multiple sports disciplines [19], in various fields, such as running [20], cycling [21], and swimming [22]. To date, some functional protocols have been published in speed skating for movement evaluation both in linear and curved motion [23], on road surfaces, on treadmills [24], and on preventive, conditional, and educational preparatory factors [25,26]. This new frontier of study has allowed for not only an increase in all the reference parameters of the last few decades but also the monitoring of the athlete’s physical health, highlighting when the load is excessive compared to the subject’s capabilities [27]. The analysis focuses on speed roller skaters, whose motor demands are very challenging in terms of mechanical stress, also due to the asymmetries of the discipline [28]. The data collected refer to the skating of two young right-handed cadet athletes and a left-handed adult male novice, compared to the performance model derived from the analysis of a right-handed elite athlete, a world champion in the same discipline. The kinematic analysis not only provides reference data useful for comparisons with the other recruited subjects but also allows for a comparison between the right and left sides in both kinematics and electromyographic findings. Moreover, it may be useful to investigate the state of muscle fatigue and its asymmetries. This study proposes useful reflections for assessing the sectional load during skating and in preventing “over-use”, as well as considering the asymmetric factor as a preventive indicator of potential damage [29]. From this consideration, the numerical and graphical results have highlighted how the recruited subjects in this study present physiological differences in both kinematic and electromyographic terms compared to the performance model. It is revealed that the technical progression closest to the performance model is that of cadet 1 (Figure 1b) as, both in percentage terms and at the accelerative level (particularly latero-medial), it overlaps with that of the elite athlete. This is less evident with cadet 2, as both the “propulsion” and “recovery” phases are more displaced compared to the model, not to mention the accelerative values and graphical indications. 

### 4.2. Electromyographic Comparisons

At the level of electromyographic acquisitions, cadet 2, particularly for the gluteus medius, showed greater conditional similarity to the performance model, slightly displaced compared to cadet 1, which presents values further from the model for the same area. The main differences are in the graphical trend, where cadet 1 seems not to fully exploit the abduction action of the gluteus medius for the entire propulsive action, anticipating and speeding up the lateral movement of the leg (Figure 2a,b). Meanwhile, the muscular involvement of the adductor longus shows that both cadet 1 and cadet 2 present trends more distant from the elite athlete in the propulsion phase. In the analysis of the respective co-activations, Figure 3 presents comparison values on the ability of muscle coordination between agonist and antagonist muscles. The co-activation value comparison indicates that the performance model maintains a balance between agonist–antagonist muscle activation throughout the recovery phase, unlike what happens with the two cadets. This is confirmed by the fact that, while the elite athlete shows high but constant values during the propulsive action, the two cadets present different and not entirely functional values. This last reference is observed from the graphical percentage trend of co-activation, which shows peaks, representing ineffective control between the muscle areas during the skating cycle. A particularly interesting comparison is made between the performance model and the novice. Recent studies have shown how kinematic and electromyographic analysis on elite athletes and novices has provided useful findings to apply in conditioning populations entering the skating discipline [26]. The kinematic analysis allows us to highlight potential limitations in the novice’s technical gesture in relation to the performance model. Despite these considerations and various discrepancies in some aspects, the graphical findings seem to show good skating homogeneity, particularly in the stability of the body in the accelerations of the latero-medial axis. The acquisition of the novice’s gluteus medius indicates that muscle activation signals are lower than those of the elite athlete, both in the propulsion and recovery phases. This result is expected but allows technical trainers in athletic preparation and prevention to use concrete data on which to base preparatory work and conditioning for any population being trained. In co-activation values, while the athlete shows a wide homogeneity in the propulsive phase, the novice presents opposite trends with peaks and declines throughout the phase (already seen in the cadet analysis, particularly cadet 2). This indicates that, despite the numerical data being close to that of the elite athlete, the graphical dynamic shows very discontinuous and poorly functional agonist-antagonist coordination, with a justified value of neuromuscular stiffness inefficiency. 

### 4.3. Asymmetry and Biomechanical Considerations

It is important to consider that, given the asymmetrical demands of the discipline, the right leg seems to be the most neuromuscularly coordinated, as it is essential for pushing curves. The novice’s asymmetry suggests that such asymmetries may mainly be induced by the inability to control the technical gesture, as the more pronounced value presented by the vastus lateralis leads us to think that the novice does not fully utilize the primary muscles for pushing, resulting in excessive compensation by accessory muscles. Biomechanical considerations, already supported by the literature [30], have shown that the activity of the gluteus maximus and gluteus medius is more pronounced in the propulsive phase, as both are agonists in the propulsive action. The graphical trend of skating linked to muscle activation is a crucial point to consider as it allows us to monitor how effectively a muscle area, in relation to functional anatomy, is activated during the cycle. Discordant values indicate alterations in skating control. This is consistent with other works proposed on ice, favoring the application of the functional protocol from which this study is derived [31]. Paying careful attention to the muscles examined in the acquisitions, it can be seen how the elite athlete and the other three recruited subjects show discordant values in the established comparisons. This highlights how the use of the protocol serves to emphasize these differences in educational terms, allowing us to investigate how this heterogeneity among the study populations is evident for a future educational work plan. The relationship between overload and injury risk has been widely recognized as an epidemiological indicator of potential injury [32], providing data in terms of prevention. Skating is considered a sport with high mechanical stress impact. It is crucial to investigate which muscle areas are most at risk of injury. Scientific research has widely confirmed that the most at-risk area for locomotor system injury in this sport is the adductor. It is, therefore, essential to investigate the potential load that this area must endure during skating. For this reason, this experimental study investigated the electromyographic activity and relative % MVIC to monitor the load state. The results showed that the elite athlete utilizes a pre-tensioning of the adductor during the right leg propulsion phase that is not excessive but should present a state of physiological activation. This pre-tensioning is favorable for the subsequent recovery phase but with a potential risk of overload injury that must be considered. Further analysis is given to the direct antagonist of the adductor longus, the gluteus medius, which is crucial for skating advancement but should not show excessive asymmetry values compared to the adductor longus. This is because it is known that a dismetry in muscle activation between the right and left adductors (80% reduction in impulse) and their respective co-activations with the gluteus medius in terms of muscle tension can make the adductor longus more susceptible to injury risk [13]. Asymmetries between muscle areas are a crucial point for injury prevention in multiple cyclic sports, where they have been specifically investigated. This concept highlights how, given the asymmetrical demands of the discipline due to counterclockwise laps, the right leg appears to be the most neuromuscular coordinated, as it is essential for pushing curves [23]. The important asymmetry indication in the novice suggests that such values may mainly be induced by an inability to control the technical gesture, particularly in terms of co-activation percentage, as the more pronounced value presented by the vastus lateralis suggests that the novice does not fully utilize the primary muscles for pushing, creating excessive compensation in the accessory muscles.

## 5. Conclusions and Limitations

This experimental study provided useful results for allowing a trainer to present objective data on which to base the conditioning of their athlete, whether a competitive athlete or a novice taking up roller skating. Kinematic analysis was not the only focus of this study, as electromyographic acquisitions also sought to evaluate the EMG peaks with their respective % MVIC and co-activations of agonist–antagonist muscles. The work described above has some strengths and limitations: the main strength is the possibility of having objective kinematic and electromyographic data for the trainer to customize the work on each athlete. The main limitation is the small sample, so that analyses of more subjects will be needed to provide robust data to support the reported results. A further limitation of this study was the different ages of the subjects studied (from 13 to 49 years). The subjects studied presented different ages and technical characteristics, useful to investigate the potential of this analysis tool in different contexts of athletic maturity. One of the two cadets was considered by the coach to be more technically gifted and one more mature from a muscular point of view, while the senior athlete came from another sport (running in the middle distance in the past) and, therefore, less likely to present muscular asymmetries and with a lower technical level than the two cadets. This allowed us, even with a small sample of subjects, to test the system’s ability to detect differences that had some correspondence with the real technical and muscular abilities of the athletes, evaluated by the coach, to be confirmed in future studies. Finally, the fact that an athlete is left-handed opens the door to potential future considerations on the role of mankinism in asymmetric disciplines (roller speed skaters always run in the counterclockwise direction with different load on left and right leg) whether it can be a factor of help or obstacle for the best performances. The methodological approach described can potentially be applied to other sports in which there is a component of asymmetry or high mechanical stress since it is possible to modify the muscles investigated and select the muscle pairs for the analysis of co-activations based on the needs of the specific sports discipline. Acceleration can also be declined on the Cartesian axis of greatest interest and on the body district of greatest interest by appropriately positioning the accelerometer. As final practical recommendations for future studies, on roller speed skating as well as on other disciplines, we suggest cooperation between physiotherapists and kinesiologists, since this type of protocol can be useful to both professionals if appropriately declined, for profitable and mutual benefit.

## Figures and Tables

**Figure 1 sensors-24-07617-f001:**
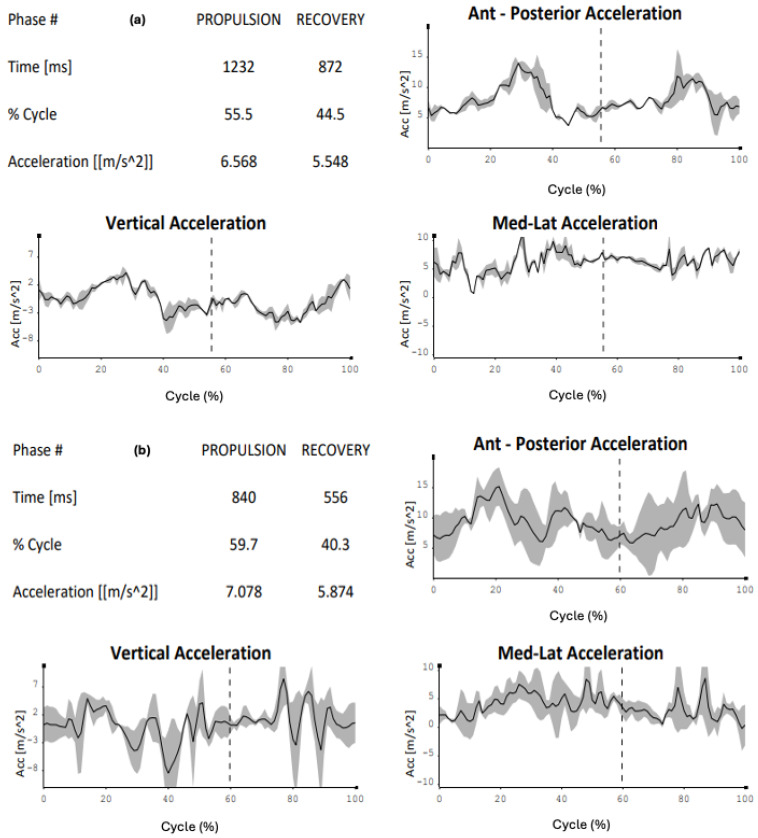
(**a**–**c**) Kinematic comparison of the right side of the elite athlete (**a**)—cadet 1 (**b**)—cadet 2 (**c**). Graphs show the acceleration for each subject in the three Cartesian axes (antero-posterior, vertical, medial–lateral) in the propulsive phase and recovery phase of the cycle, separated by the dotted vertical line; the tables indicate, for each subject, the time of the propulsive phase and the recovery phase, the percentage duration of the propulsion phase and recovery phase of the skating cycle and the average value of the medial–lateral acceleration.

**Figure 2 sensors-24-07617-f002:**
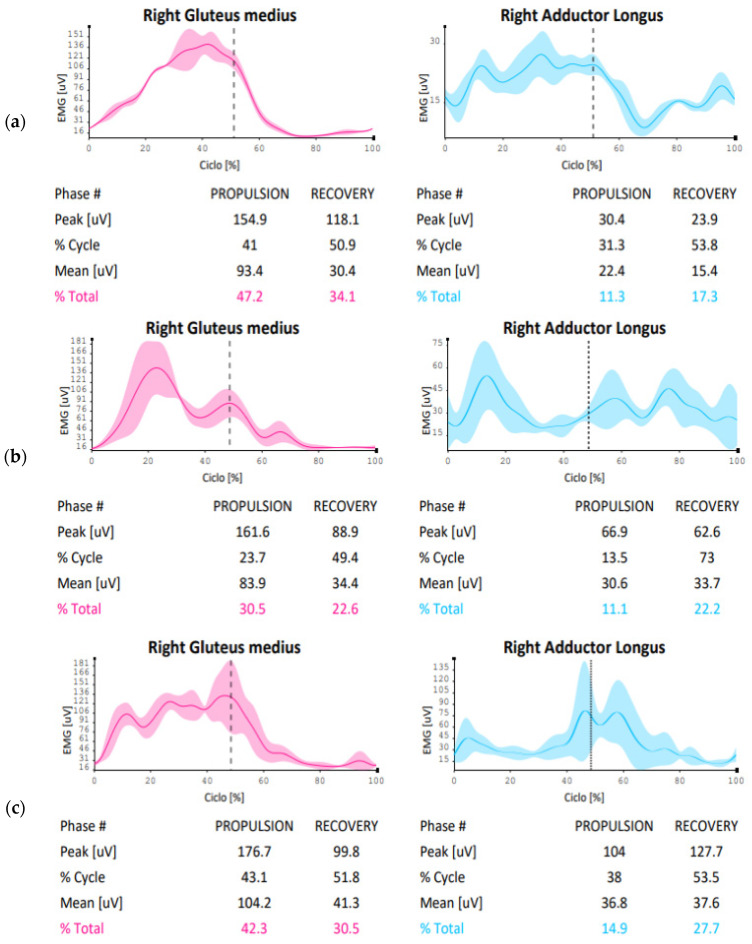
(**a**–**c**) EMG comparison of the right side of the elite athlete (**a**)—cadet 1 (**b**)—cadet 2 (**c**).

**Figure 3 sensors-24-07617-f003:**
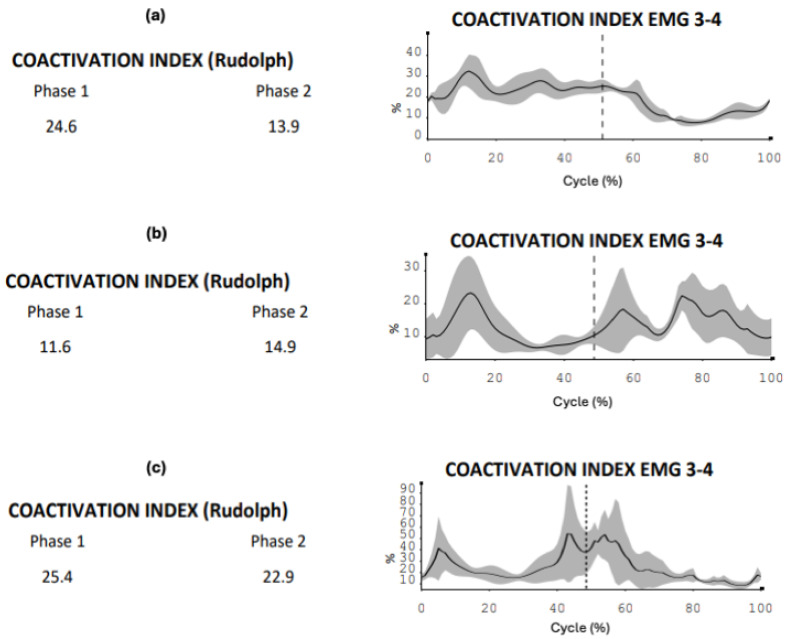
(**a**–**c**) Co-activation comparison of the right side (gluteus medius-adductor) elite athlete (**a**)—cadet 1 (**b**)—cadet 2 (**c**). The figure indicates on the left side the average co-activation value of the two muscles investigated (Rudolph index, expressed in percentage terms) during phase 1 (propulsive) and phase 2 (recovery) of the skating cycle in the three subjects ((**a**): first line, (**b**): second line, (**c**): third line); the right side of the figure graphically indicates the distribution of co-activations over time (skating cycle): the two phases of propulsion and recovery are separated by a vertical dotted line.

**Figure 4 sensors-24-07617-f004:**
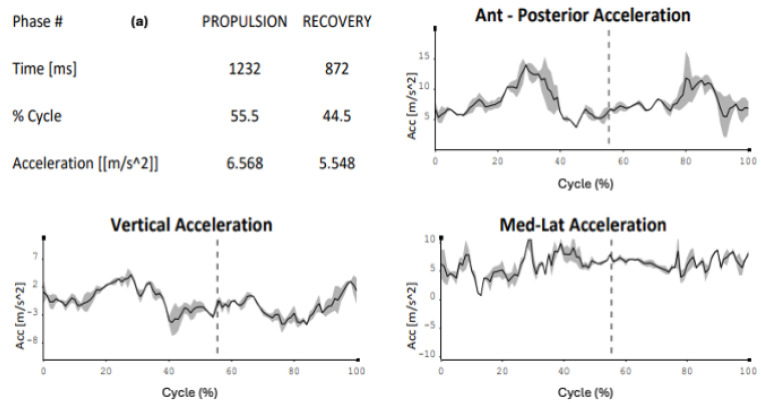
(**a**,**b**) Kinematic comparison of the right side of the elite athlete (**a**) and left side of novice (**b**). Graphs show the acceleration for each subject in the three Cartesian axes (antero-posterior, vertical, medial–lateral) in the propulsive phase and recovery phase of the cycle, separated by the dotted vertical line; the tables indicate, for each subject, the time of the propulsive phase and of the recovery phase, the percentage duration of the propulsion phase and recovery phase of the skating cycle and the average value of the medial–lateral acceleration.

**Figure 5 sensors-24-07617-f005:**
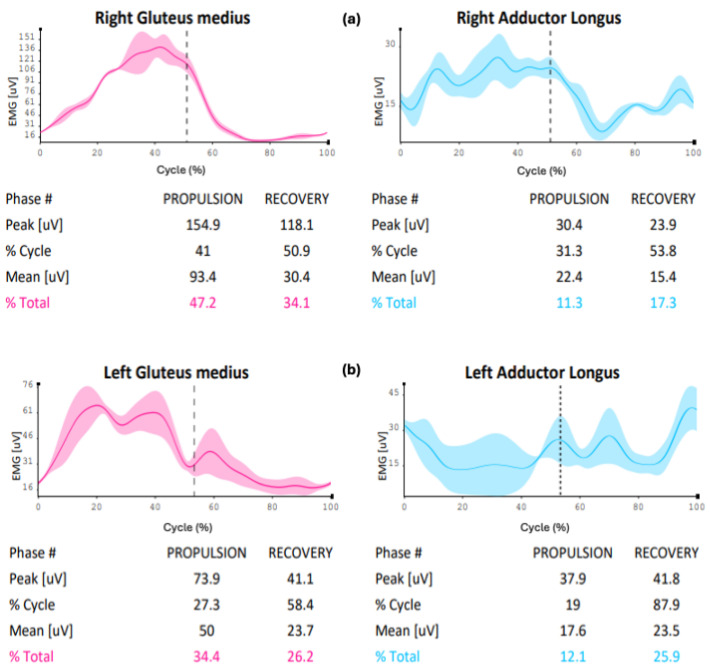
(**a**,**b**) EMG comparison of the right side of the elite athlete (**a**) and the left side of the novice (**b**). The figure indicates the sEMG value during the skating cycle phase of the right gluteus medius and right adductor longus muscles, in the two subjects; the table below each graph indicates, during the propulsion and recovery phase (separated by the dotted vertical line), the peak electromyographic value, the percentage of the cycle at which this peak develops, the average electromyographic value, its percentage distribution in the two phases.

**Figure 6 sensors-24-07617-f006:**
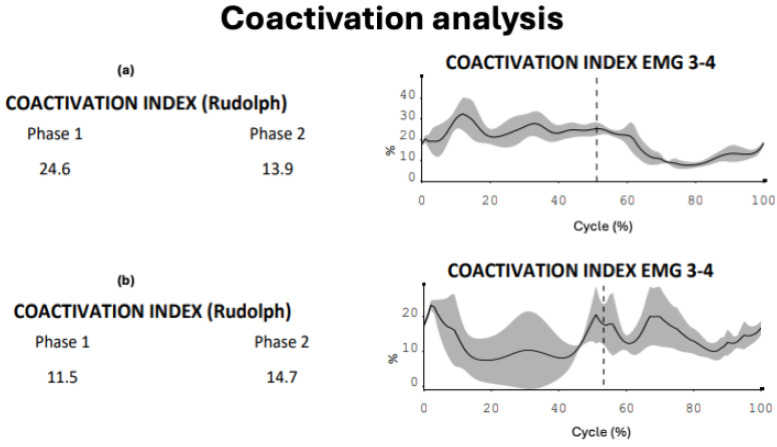
(**a**,**b**) Co-activation comparison of the right side of the elite athlete (**a**) and the left side of the novice (**b**). The figure indicates on the left side the average co-activation value of the two muscles investigated (Rudolph index, expressed in percentage terms) during phase 1 (propulsive) and phase 2 (recovery) of the skating cycle in the two subjects ((**a**): first line, (**b**): second line,); the right side of the figure graphically indicates the distribution of co-activations over time (skating cycle): the two phases of propulsion and recovery are separated by a vertical dotted line.

**Table 1 sensors-24-07617-t001:** Asymmetry index analysis for cadet 1. The table shows, for the vastus lateralis, biceps femoris, gluteus medius and adductor longus muscles, the right (dx.) and left (sx.) electromyographic values of the propulsive and recovery phase; it also shows the average percentage co-activation values of these muscles in two pairs (pair one vastus lateralis-biceps femoris muscles, pair two gluteus medius-adductor longus muscles).

ANALYSIS MVIC % AND CO-ACTIVATIONS (Rudolph Index)		
Districts	Analysis dx. 1 (Propulsion-Recovery)	Analysis sx. 1
Vastus lateralis	229.1–152.4 µV	310.1–267.2 µV
Biceps femoralis	21.9–30.3 µV	29.1–43.7 µV
Co-activations	14.4–18.7%	20.8–28.4%
**Districts**	**Analysis dx. 2 (propulsion-recovery)**	**Analysis sx. 2**
Medius gluteus	40.9–51 µV	35.1–22.2 µV
Adductor longus	46.8–66.4 µV	85–106 µV
Co-activations	29.1–22.9%	29.6–19.1%

**Table 2 sensors-24-07617-t002:** Asymmetry index analysis for cadet 2. The table shows, for the vastus lateralis, biceps femoris, gluteus medius and adductor longus muscles, the right (dx.) and left (sx.) electromyographic values of the propulsive and recovery phase; it also shows the average percentage co-activation values of these muscles in two pairs (pair one vastus lateralis-biceps femoris muscles, pair two gluteus medius-adductor longus muscles).

ANALYSIS MVIC % AND CO-ACTIVATIONS (Rudolph Index)		
Districts	Analysis dx. 1 (Propulsion-Recovery)	Analysis sx. 1
Vastus lateralis	90.4–128.1 µV	194–212.9 µV
Biceps femoralis	36.6–33 µV	30.8–32.3 µV
Co-activations	33.9–17.5%	26.4–20.6%
**Districts**	**Analysis dx. 2 (propulsion-recovery)**	**Analysis sx. 2**
Medius gluteus	80.1–56.1 µV	108–86.5 µV
Adductor longus	42.8–111.2 µV	49.8–125.8 µV
Co-activations	24.2–27.1%	33.3–25.4%

**Table 3 sensors-24-07617-t003:** Asymmetry index analysis for the novice. The table shows, for the vastus lateralis, biceps femoris, gluteus medius and adductor longus muscles, the right (dx.) and left (sx.) electromyographic values of the propulsive and recovery phase; it also shows the average percentage co-activation values of these muscles in two pairs (pair one vastus lateralis-biceps femoris muscles, pair two gluteus medius-adductor longus muscles).

ANALYSIS MVIC % AND CO-ACTIVATIONS (Rudolph Index)		
Districts	Analysis dx. 1 (Propulsion-Recovery)	Analysis sx. 1
Vastus lateralis	378.3–398.7 µV	74.3–54.9 µV
Biceps femoralis	19.7–28.4 µV	19.8–17.8 µV
Co-activations	13–17.7%	16.2–12%
**Districts**	**Analysis dx. 2 (propulsion-recovery)**	**Analysis sx. 2**
Medius gluteus	27.3–10.2 µV	13.2–9.9 µV
Adductor longus	41.7–38.8 µV	45.8–53.2 µV
Co-activations	23.8–8%	6.6–4.6%

## Data Availability

Data may be obtained from the corresponding author upon reasonable request.

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
