# Peer review of "Training in Roller Speed Skating: Proposal of Surface Electromyography and Kinematics Data for Educational Purposes in Junior and Senior Athletes"

_sensors, 2024, doi:10.3390/s24237617_

Round 1

Reviewer 1 Report

Comments and Suggestions for Authors

The authors carried out a study that sought to investigate and analyse data from electromyographic and triaxial accelerometric  patterns between a roller skater novice and two cadets. I found the content more than suitable for the journal. The study is correctly designed and technically sound.

The authors generally did a good job, however the fact that they only had three roller speed skaters for analysis is a very reducing point of the work. However, I appreciate the contribution of the author/s to focus on the  area where the specific data are still missing, even if be of interest only to a limited number of people.

Below you can find in detail my comments that may be useful in helping the work reach its full potential.

Results

The data and analyses are presented appropriately.

Line 131: Change to Elite athlete vs. cadets 1-2 analysis (as subtitle):

In general I suggest always start with the text and then insert the particular figure or table.

Line 162: change to Elite athlete vs. novice analysis (as subtitle)

The same suggestion as above: always start with the text an then insert the figure or table.

All figures in general:

must show clearly which one is a -  b - c  part.

- designation of the axis of the graph should be corrected: Cycle and not Ciclo / Cicle as it is now in most of graphs. The same error is shown in

 Discussion

Well written , however, for a reader would be better to divide in paragraphs regarding the key points.

It's   essential   to recognize    that    these    are    preliminary findings    that    require    further    in-depth examination  in  the  future,  considering  the recommendations  outlined  in  the  research limitations section.

The biggest criticism is actually the fact that only three athletes were studied. Still, I believe that a study of this genesis has immense practical applicability from the coach's perspective and this is not reflected in the text. In fact, I consider that perhaps adding a specific point at the end, properly structured that provides practical recommendations would add value to the study.

Author Response

Dear reviewer,
thank you for reading the manuscript and suggesting us how to improve it. We appreciated every correction and we agree with all of them. We have inserted the requested changes directly into the text, highlighting them for your convenience with the green color, to be able to distinguish them from the changes requested by the other reviewers for which we have used other colors. Hoping to have made the text acceptable for publication, we send our best regards.
Luca Miceli and the other co-authors

Reviewer 2 Report

Comments and Suggestions for Authors

I consider it an interesting work, but I think there are details to be modified that would improve it:

1) On line 82, the electromyographic and kinematic protocol previously created by you (the authors of this new study) is mentioned for the first time, giving a reference [3]. Then on line 92, it is said that said protocol is published and validated. It is true that it is published, but the VALIDATION in my opinion is in a very preliminary phase, since said study still has almost no citations (in WoS it currently has only 8 citations and all 8 are self-citations). In addition, there are international standards for validation (ISO 15189, Good Laboratory Practices, FDA Guidances, etc.). I think that using the word VALIDATION in this case is incorrect.

2) On line 100, where the Seniam Project is cited, the citation is from the year 2000 [18], perhaps it would be convenient to add some more recent citation in relation to its use and utility.

 3) In the “Results” section and regarding the Figures, I think they would improve if the information in the figure caption were expanded. In this regard, let me point out that the general rules for scientific papers indicate that, both in tables and figures, the titles of the tables must contain the information necessary to correctly interpret the table or figure without resorting to the text. This fundamentally allows non-expert readers to better interpret what the table summarizes and, in my opinion, would improve the quality of the work. I had trouble understanding several of the tables without thoroughly reviewing the text.

 4) In lines 42, 144, 167, 181, 222, 225, 259, 267, 285, 301 the word “significance” is used. I understand that statistics have not been used in this pilot study (given the small sample), so using this term, in some cases, can be risky, since statistical significance can be implied.

 5) The conclusions speak of the limitations of the study (line 346) and, in my opinion, the difference in age between the different subjects (13, 49 and 31 years) is a relevant limitation, so it should be added to the text.

Author Response

Dear reviewer,
thank you for reading the manuscript and suggesting how to improve it. We appreciated every correction and we agree with all of them. We have inserted the requested changes directly into the text, highlighting them for your convenience with the red color, to be able to distinguish them from the changes requested by the other reviewers for which we have used other colors. Hoping to have made the text acceptable for publication, we send our best regards.
Luca Miceli and the other co-authors

Reviewer 3 Report

Comments and Suggestions for Authors

This study explores the use of surface electromyography (sEMG) and kinematic analysis to assess and improve the technical skills and injury prevention strategies of roller speed skaters. The researchers conducted a pilot study involving three participants—two junior athletes and a novice—comparing their performance with that of an elite roller speed skater. sEMG and triaxial accelerometers were used to monitor muscle activity and movement patterns in the lower limbs during the propulsion and recovery phases of skating. The findings highlight notable differences in muscle activation, co-activation, and kinematic patterns between athletes of varying skill levels. These insights aim to aid in the development of personalized training regimens and offer data-driven guidance to enhance performance and minimize injury risks for athletes at different skill levels. The study underscores the potential of sEMG and kinematic analysis as educational tools for conditioning and coaching in sports that involve high mechanical stress and asymmetrical movement patterns.

Suggested improvements

1.The background information in the introduction is well-detailed but could benefit from a clearer focus on the specific gaps this study addresses. Emphasize the novelty of applying sEMG and kinematic analysis to roller speed skating for educational purposes and differentiate it from prior research.

2.Consider briefly mentioning the applicability of this approach to other sports involving asymmetrical movements or high mechanical stress.

3.Clarify the rationale behind selecting the three study subjects and discuss how the small sample size might affect the generalizability of the results.

Author Response

Dear reviewer,
thank you for reading the manuscript and suggesting how to improve it. We appreciated every correction and we agree with all of them. We have inserted the requested changes directly into the text, highlighting them for your convenience with the blue color, to be able to distinguish them from the changes requested by the other reviewers for which we have used other colors. Hoping to have made the text acceptable for publication, we send our best regards.
Luca Miceli and the other co-authors
